# Prevalence and Epidemiological and Clinical Features of Bacterial Infections in a Large Cohort of Patients Hospitalized for COVID-19 in Southern Italy: A Multicenter Study

**DOI:** 10.3390/antibiotics12071124

**Published:** 2023-06-29

**Authors:** Lorenzo Onorato, Federica Calò, Paolo Maggi, Enrico Allegorico, Ivan Gentile, Vincenzo Sangiovanni, Vincenzo Esposito, Chiara Dell’Isola, Giosuele Calabria, Raffaella Pisapia, Angelo Salomone Megna, Alfonso Masullo, Elio Manzillo, Grazia Russo, Roberto Parrella, Giuseppina Dell’Aquila, Michele Gambardella, Felice Di Perna, Mariantonietta Pisaturo, Nicola Coppola

**Affiliations:** 1Infectious Diseases Section, Department of Mental Health and Public Medicine, University of Campania “L. Vanvitelli”, 80138 Naples, Italymariantonietta.pisaturo@unicampania.it (M.P.); nicola.coppola@unicampania.it (N.C.); 2Infectious Disease Unit, A.O. S Anna e S Sebastiano Caserta, 81100 Caserta, Italy; paolo.maggi@unicampania.it; 3Emergency Unit, PO Santa Maria delle Grazie, 80078 Pozzuoli, Italy; enrico.allegorico@aslnapoli2nord.it; 4Infectious Disease Unit, University Federico II, 80138 Naples, Italy; ivan.gentile@unina.it; 5Third Infectious Disease Unit, AORN dei Colli, PO Cotugno, 80131 Naples, Italy; sangio.vincenzo@gmail.com; 6IV Infectious Disease Unit, AORN dei Colli, PO Cotugno, 80131 Naples, Italy; vincenzoesposito@ospedalideicolli.it; 7Hepatic Infectious Disease Unit, AORN dei Colli, PO Cotugno, 80131 Naples, Italy; 8IX Infectious Disease Unit, AORN dei Colli, PO Cotugno, 80131 Naples, Italy; g.calabria@tin.it; 9First Infectious Disease Unit, AORN dei Colli, PO Cotugno, 80131 Naples, Italy; raffipisapia@gmail.com; 10Infectious Disease Unit, A.O. San Pio, PO Rummo, 82100 Benevento, Italy; 11Infectious Disease Unit, A.O. San Giovanni di Dio e Ruggi D’Aragona Salerno, 84131 Salerno, Italy; al.masullo@alice.it; 12VIII Infectious Disease Unit, AORN dei Colli, PO Cotugno, 80131 Naples, Italy; manzillo@libero.it; 13Infectious Disease Unit, Ospedale Maria S.S. Addolorata di Eboli, ASL Salerno, 84131 Salerno, Italy; gr.russo@aslsalerno.it; 14Respiratory Infectious Diseases Unit, AORN dei Colli, PO Cotugno, 80131 Naples, Italy; roberto.parrella@ospedalideicolli.it; 15Infectiou Disease Unit, A.O. Avellino, 83100 Avellino, Italy; dellaquilagiuseppina@libero.it; 16Infectious Disease Unit, P.O. S. Luca, ASL Salerno, 84078 Vallo della Lucania, Italy; gambardella1960@gmail.com; 17Pneumology Unit, AORN Caserta, 81100 Caserta, Italy; felice.diperna@ospedale.caserta.it

**Keywords:** COVID-19, bacterial infections, superinfections, antimicrobial prescriptions, PCT

## Abstract

Background: The aim of this study was to evaluate the prevalence of bacterial infections and antimicrobial prescriptions in a large cohort of COVID-19 patients and to identify the independent predictors of infection and antibiotic prescription. Methods: All consecutive patients hospitalized for COVID-19 from March 2020 to May 2021 at 1 of the 17 centers participating in the study were included. All subjects showing a clinical presentation consistent with a bacterial infection with microbiological confirmation (documented infection), and/or a procalcitonin value >1 ng/mL (suspected infection) were considered as having a coinfection (if present at admission) or a superinfection (if acquired after at least 48 h of hospital stay). Results: During the study period, of the 1993 patients, 42 (2.1%) presented with a microbiologically documented infection, including 17 coinfections and 25 superinfections, and 267 (13.2%) a suspected infection. A total of 478 subjects (24.5%) received an antibacterial treatment other than macrolides. No independent predictors of confirmed or suspected bacterial infection were identified. On the contrary, being hospitalized during the second wave of the pandemic (OR 1.35, 95% CI 1.18–1.97, *p* = 0.001), having a SOFA score ≥3 (OR 2.05, 95% CI 1.53–2.75, *p* < 0.001), a severe or critical disease (OR 1.66, 95% CI 1.24–2.23, *p* < 0.001), and a high white blood cell count (OR 1.03, 95% CI 1.004–1.06, *p* = 0.023) were all independently related to having received an antimicrobial prescription. Conclusions: Our study reported a high rate of antimicrobial prescriptions despite a limited number of documented or suspected bacterial infections among the large cohort of hospitalized COVID-19 patients.

## 1. Introduction

After more than two years, the COVID-19 pandemic is still a global health threat, with a total of 765 million cases and 6.9 million deaths occurring up to April 2023 [1]. Although the availability of several safe vaccine options has significantly reduced the morbidity and mortality related to SARS-CoV-2 infection [2], effective therapeutic strategies, particularly for deeply immunocompromised patients, are still needed. 

It has been clearly described that the pandemic, particularly during the first waves, has deeply impacted almost every aspect of global healthcare, such as the rate of hospital admission for several diseases. Indeed, many studies reported during the first months of the COVID-19 pandemic a dramatic fall in the number of patients admitted for several clinical conditions, including myocardial infarction, sepsis, or chronic obstructive pulmonary disease [3,4]. The significant changes in clinical characteristics of patients admitted and the need for a dramatic and sudden reorganization of the facilities have obviously impacted several aspects of healthcare, including the prescribing behavior of physicians. 

Although the available data estimate a 5% prevalence of co-infection and 16% prevalence of superinfection, more than 65% of patients hospitalized for COVID-19 [5] received an antimicrobial treatment, especially during the first waves of the pandemic [6]. In fact, because of the lack of knowledge regarding the clinical history of the disease and the difficulties in the diagnostics of superinfections, due to the overlap in clinical, biochemical, and radiological presentation between severe bacterial and viral infections, most physicians were induced to overprescribe antibiotic treatments, often without clear evidence of bacterial infections [7]. Moreover, the difficulties in applying correct infection control practices, the overcrowding of facilities, and the shortages of adequately trained staff have led to an increase in healthcare-associated infections in COVID-19 settings, causing further antibiotic consumption.

Italy is one of the countries with the highest rates of antibiotic consumption and antimicrobial resistance in Europe [8], with a significant burden in terms of morbidity and mortality [9]. Thus, the monitoring and rate and appropriateness of antimicrobial prescriptions are of utmost importance in our area to limit the spread of resistant pathogens. In the present study, we collected the data of a large cohort of patients admitted for COVID-19 from March 2020 to May 2021 in 17 healthcare facilities in the Campania region, in southern Italy. We aimed at evaluating the prevalence of bacterial infection and antimicrobial prescriptions in this setting, and to identify the independent predictors of both infections and prescriptions. 

## 2. Methods

We conducted a retrospective multicenter cohort study; all hospitalized patients admitted for COVID-19 to 1 of the 17 participating centers from March 2020 to May 2021 were evaluated for enrolment. The clinical centers were located in different cities of the Campania region in southern Italy; 15 out of 17 were Infectious Disease Units, while the remaining 2 were sub-intensive care Units; all these centers have collaborated in previous studies [10,11]. 

The patients were adults (≥18 years), hospitalized with a diagnosis of SARS-CoV-2 infection confirmed by a positive reverse transcriptase-polymerase chain reaction (RT-PCR) on a naso-oropharyngeal swab. The study period was from 28 February 2020 to 31 May 2021 [10,11]. Patients were excluded if they were <18 years old or did not provide informed consent to participate in the study. For each patient included we collected demographic, clinical, and microbiological data, including age, gender, date, and ward of admission, length of hospital stay, presence of comorbidities, Charlson Comorbidity Index (CCI) [12], which has been related to COVID-19 outcomes in several studies [13,14], severity of COVID-19 according to the WHO definitions [15], Sequential Organ Failure Assessment (SOFA) Score, a well-known severity index score used in the identification of sepsis according to the Sepsis-3 definition [16], baseline laboratory parameters, clinical isolates, and treatment received (supplemental oxygen therapy, antimicrobial, and corticosteroid treatment). 

### 2.1. Definitions

Infections were defined according to the CDC/NHSN Surveillance Definitions 2023 [17]. Furthermore, patients with signs or symptoms suggestive of superinfection and a procalcitonin (PCT) value > 1 ng/mL, but without a microbiologically documented infection, were considered as having a suspected bacterial infection.

A coinfection was defined as a microbiologically documented bacterial infection diagnosed within 48 h after hospital admission, while infections diagnosed after 48 h of hospital stay were considered superinfections. 

We identified three waves of the pandemic: the first wave included patients admitted from March to August 2020, the second was those hospitalized from September 2020 to February 2021, while the patients admitted from March 2021 onwards were included in the third wave. 

### 2.2. Outcomes

The outcomes evaluated included the prevalence of definite or suspected bacterial infections, as previously defined, and the prevalence of antimicrobial prescriptions. Furthermore, we aimed at identifying independent predictors of bacterial infections, as well as of antibiotic prescriptions. Since macrolides, particularly during the first wave of the pandemic, were prescribed for reasons not strictly related to their antimicrobial activity, this class was excluded from the analysis of antibiotic prescription. 

### 2.3. Statistical Analysis

Continuous variables were summarized as mean and standard deviation, or median and interquartile range, and categorical variables as absolute and relative frequencies. For continuous variables, the differences were evaluated by Student’s *t*-test or Mann–Whitney U test, as indicated; categorical variables were compared by the chi-squared test, using exact procedures if needed. To identify independent predictors of bacterial infections and of antimicrobial prescriptions, a multivariate analysis using logistic regression was performed; clinically meaningful variables, and variables associated with the evaluated outcomes at univariate analysis with a *p* value < 0.05 were included in the model. A *p* value < 0.05 was considered statistically significant. The analysis was performed using IBM SPSS v 21.0 (Armonk, NY, USA).

### 2.4. Ethics Statement

The study was approved by the Ethics Committee of the University of Campania L. Vanvitelli, Naples (n°10877/2020). All procedures performed in this study were in accordance with the ethical standards of the institutional and/or national research committee and with the 1964 Helsinki Declaration and its later amendments or comparable ethical standards. Informed consent was obtained from all participants included in the study.

## 3. Results

### 3.1. Baseline Characteristics of Patients

During the study period, a total of 1993 patients were enrolled, with a mean age of 62.2 ± 16 years; 61.5% of subjects were males. 

The demographic, clinical, and biochemical characteristics of the patients included are shown in Table 1. The most common comorbidities included cardiovascular diseases in 556 (27.9%) cases, diabetes in 406 (20.4%), and chronic obstructive pulmonary disease in 208 (10.4%) cases; the median CCI was three (IQR: 3). Regarding the severity of the outcome of COVID-19 during hospitalization, 784 (39.3%) patients presented with a mild or moderate disease, 348 (17.5%) with a severe, and 577 (28.9%) with a critical outcome, while the median SOFA score was 2 (IQR: 3). Approximately one-third of patients did not need supplemental oxygen therapy during hospitalization, 41.3% were treated with a nasal cannula or simple face mask, 8.0% required a high-flow nasal cannula (HFNC), 18.6% received continuous positive airway pressure (CPAP) or non-invasive ventilation (NIV), 0.9% required mechanical ventilation, 72.1% of patients were treated with corticosteroids, and the overall hospital mortality was 10.4%. 

### 3.2. Bacterial Infections and Antimicrobial Prescriptions

A total of 17 (0.9%) out of the 1993 patients enrolled presented with a microbiologically documented bacterial infection at admission, while 25 (1.2%) acquired a bacterial infection during the hospital stay. The most common sites of infection included primary or CVC-related bloodstream infections (23.8% of cases) and urinary tract infections (35.7% of cases). Among Gram-positive bacteria, *S. aureus* (16.7%) and *Enterococcus* spp. (21.4%) were commonly isolated strains, while *Enterobacterales* accounted for 70% of all isolated Gram-negative bacteria. 

Additionally, 267 (13.4%) patients had a procalcitonin value ≥ 1 and were classified as having a suspected bacterial infection. 

Despite these findings, that is, a definite or suspected bacterial infection in 15.5% of cases, 416 patients (20.8%) received treatment with a macrolide and 478 (24%) with an antimicrobial of a different class. 

### 3.3. Demographic, Clinical, and Biochemical Characteristics of Patients with or without Bacterial Infections

As shown in Table 2, patients with confirmed or suspected bacterial infection showed a higher rate of comorbidities, such as cardiovascular diseases (31.7 vs. 27.2%, *p* = 0.028), diabetes (25.6 vs. 19.4%, *p* = 0.006), chronic kidney diseases (13.9 vs. 7.7%, *p* < 0.001), chronic liver diseases (7.4 vs. 2.4%, *p* < 0.001), and HIV infection (1.3 vs. 0.5%, *p* < 0.001). Regarding the biochemical features, a higher baseline creatinine value was observed among patients with bacterial infection (*p* = 0.001). The modalities of oxygen treatment were differently distributed in the groups (*p* = 0.001); in particular, a higher proportion of patients in the bacterial infection group received treatment with a nasal cannula or simple face mask (59.7 vs. 48.4%), while a lower proportion needed HFNC, CPAP, or NIV (22 vs. 34.9%, *p* < 0.001). Subjects with bacterial infections obviously received antibacterial treatments more frequently (*p* = 0.001); moreover, a higher rate of these patients was treated with corticosteroids (88.3% vs. 69.1%, *p* < 0.001), although no significant difference was observed in the prevalence of severe or critical disease (*p* = 0.19). No difference in hospital mortality was observed in the groups.

### 3.4. Demographic, Clinical, and Biochemical Characteristics of Patients Who Received or Did Not Receive Antimicrobial Treatment

When stratifying patients according to treatment received, we found that patients who were treated with antibiotics were older (65.2 ± 16 vs. 61.3 ± 15.9 years, *p* < 0.001), were mostly admitted during the second wave of the pandemic (68.2 vs. 49.8%, *p* < 0.001), and had a higher burden of comorbidities, as demonstrated by a higher median CCI (3, IQR:4 vs. 2, IQR:3, *p* = 0.002), and a higher prevalence of diabetes (24.5 vs. 19.1%, *p* = 0.017) and chronic kidney disease (12.3 vs. 7.5%, *p* = 0.02) (Table 3). Regarding the severity, the SOFA score was higher in the antibiotic treatment group (*p* < 0.001) as well as in the proportion of patients with severe or critical disease (65.1% vs. 49.9%, *p* < 0.001) and the rate of subjects requiring HFNC, CPAP, NIV, or mechanical ventilation (42.7% vs. 29.1%, *p* < 0.001); consequently, the prescription of corticosteroid treatment was more common in the first group (93.1 vs. 65.4%, *p* < 0.001). Finally, patients in the antibiotic treatment group presented with a higher white blood cell count at baseline (*p* < 0.001). When we focused our analysis on patients who received an antimicrobial prescription and stratified them according to the presence of a bacterial infection (Appendix A), we found that subjects without evidence of a superinfection presented with a higher prevalence of severe or critical disease (*p* = 0.008), a higher median SOFA score (*p* < 0.001) and a higher rate of oxygen therapy delivered through HFNC, CPAP, NIV, or mechanical ventilation (*p* < 0.001).

### 3.5. Independent Predictors of Bacterial Infections and Antimicrobial Treatment

In a multivariate logistic regression analysis, no independent predictors of confirmed or suspected bacterial infection were identified (Table 4). 

Regarding antibiotic treatment, being hospitalized during the second wave of the pandemic (OR 1.35, 95% CI 1.18 to 1.97, *p* = 0.001), having a SOFA score ≥ 3 (OR 2.05, 95% CI 1.53 to 2.75, *p* < 0.001), a severe or critical disease (OR 1.66, 95% CI 1.24 to 2.23, *p* < 0.001), and a higher white blood cell count (OR 1.03, 95% CI 1.004 to 1.06, *p* = 0.023) were all independently related to having received an antimicrobial prescription (Table 5).

## 4. Discussion

In the present study, we reported a 15.5% prevalence of microbiologically confirmed or suspected bacterial infections in a large cohort of COVID-19 patients hospitalized at 1 of 17 healthcare facilities, non-ICU, in the Campania region, southern Italy, during the first three waves of the pandemic. No independent predictors of infection were identified in the multivariate analysis, while having a SOFA score > 2, a severe or critical disease, and a higher white blood cell count were all independently related to having received an antimicrobial treatment.

The prevalence of bacterial infections found in our cohort is consistent with the data reported in the literature. A systematic review and meta-analysis conducted by Langford and colleagues [5], including 24 studies published during the first wave of the pandemic, reported a 3.5% prevalence of co-infection and a 14.3% prevalence of secondary infections. Although in a few reports [18], a higher rate of bacterial coinfections compared to secondary infections was observed, several papers described a nosocomial acquisition for most of the bacterial infections diagnosed among COVID-19 patients [19]. A retrospective analysis of surveillance data conducted during the first year of the pandemic among 26 hospitals in Israel [20] demonstrated a progressive reduction in the incidence of bloodstream infections after the first wave, despite a lower proportion of severe cases during this wave, suggesting that the high incidence of healthcare-associated infections was mainly related to a lack of infection prevention and control (IPC) practices, rather than to patient-related factors. Indeed, particularly during the first waves of the pandemic, the IPC behavior of healthcare workers was mainly directed at self-protection rather than to avoid cross-transmission of pathogens among patients [7]. 

Regarding the risk factors for bacterial infections, as expected, several studies in the literature showed a higher prevalence in patients admitted to ICUs. In a meta-analysis with meta-regression conducted among 171 observational studies published up to February 2021, ICU admission (OR 18.8, 95% CI 6.5–54.8) and mechanical ventilation (OR 1.41, 95% CI 1.30–1.52) were the main predictors of bacterial infection [21]. A retrospective observational study including 81 COVID-19 and 144 non-COVID-19 patients receiving invasive ventilation demonstrated a higher incidence of ventilator-associated pneumonia (VAP) in the first group; the reduced VAP-free survival was confirmed also in the adjusted analysis through the Cox regression model [22]. No independent predictors of superinfection were identified in the present analysis, although we should consider that all patients included in this study were hospitalized in medical wards, and only a limited number of them (0.9%) required ICU admission during hospital stay. A large proportion of our patients presented with significant comorbidities, including cardiovascular, metabolic, or chronic diseases, and it is well known that COVID-19 itself can cause severe injuries to several organs and systems [23,24,25]; however, no correlation between these comorbidities and bacterial infections was found in our study. 

Despite the low prevalence of microbiologically documented bacterial infections, in the present study, a significant proportion of patients received antimicrobial treatment during hospital stay. This finding is consistent with several reports in the literature. A systematic review and meta-analysis by Langford and co-workers [6], including 154 studies published from January to June 2020, reported a prevalence of antimicrobial prescriptions of 74.6% among COVID-19 patients, and of up to 86.4% in ICU patients. Our rate of prescriptions was clearly lower, but we should consider that all studies included in this meta-analysis were conducted during the first wave of the pandemic, when high diagnostic uncertainty and lack of knowledge regarding the natural history of the disease led to more inappropriateness in antimicrobial prescriptions. However, the incidence of COVID-19 in our geographical area during the first months of the pandemic was quite low compared to other regions with a small proportion of patients admitted during that period presenting with a severe disease [10], which has been demonstrated in our work to be an important driver of antibiotic use. In our cohort, we found a higher severity rate in patients who received an antimicrobial prescription despite not having any evidence of bacterial infection compared to those treated for a confirmed or suspected bacterial infection. This demonstrates the tendency of physicians, observed particularly during the first phase of the pandemic, to prescribe antibiotics to critically ill COVID-19 patients regardless of the presence of clinical and microbiological data consistent with a bacterial infection. In this perspective, having received an antimicrobial prescription represents a hallmark identifying a subset of patients with more severe disease. This observation has been reported in other papers. In particular, a retrospective multicenter analysis including 1705 hospitalized COVID-19 patients with a 3.5% prevalence of documented coinfections showed that older age, being admitted to hospital during the first months of the pandemic, and the need for supplemental oxygen therapy were independent predictors of early empiric antibiotic therapy [26]. Similarly, in a retrospective study including 1027 COVID-19 patients, among the subgroup of subjects without bacterial infections, those receiving antimicrobial therapy presented with a higher rate of mechanical ventilation at 5 days from hospital admission [27].

The main strength of our paper is the large number of patients (1993) enrolled across the first three waves of the pandemic, for most of whom several demographic and clinical data were available. Furthermore, we evaluated meaningful clinical outcomes, including clinical severity of disease and in-hospital mortality. However, there are some limitations, including the retrospective study design and the definitions used to identify patients with bacterial infections. The most important limitation of our study is the low number of microbiologically documented infections, mainly due to the well-documented difficulties of collecting and analyzing microbiological samples in such a difficult-to-manage population, particularly during the first waves [19]. The diagnostic challenge in this setting is further increased by the wide overlapping in clinical, biochemical, and radiological features between the presentation of severe COVID-19 itself and of bacterial superinfections. To overcome these difficulties and to avoid an obvious underestimation of the prevalence of bacterial infections, we decided to include in our analysis patients without a microbiologically documented infection, but with a suspected infection based on the procalcitonin (PCT) values. Many studies have demonstrated that PCT shows a significantly higher diagnostic accuracy of bacterial infection in COVID-19 patients compared to alternative biomarkers [28,29]. In particular, in a cohort study enrolling COVID-19 patients admitted to an ICU, PCT showed a 96% specificity and 93% positive predictive value of bacterial superinfection when using a cut-off of 1 ng/mL [28]. 

## 5. Conclusions

The present study reports a high rate of antimicrobial prescriptions despite a limited number of documented or suspected bacterial infections among a large cohort of hospitalized COVID-19 patients. These data underline the need to enhance antimicrobial stewardship programs during the different phases of health emergencies, such as in the pandemic, to reduce the inappropriate use of antimicrobials, the principal driver of the development of antimicrobial resistance [30].

## Figures and Tables

**Table 1 antibiotics-12-01124-t001:** Demographic, clinical, and biochemical characteristics of patients enrolled.

N° of patients	1993
Mean age (SD), years	62.2 (16)
Males, n° (%)	1226 (61.5)
Mean length of hospital stay (SD), days	16 (10.6)
Admitted to ID wards, n° (%)	1794 (90)
Waves of pandemic, n° (%) - First - Second - Third	317 (15.9) 1080 (54.2) 565 (28.3)
Charlson Comorbidity Index (median, IQR)	3 (3)
Comorbidities, n° (%) - Cardiovascular disease - Diabetes - COPD - Chronic kidney disease - Chronic liver disease - Malignancies - HIV	556 (27.9) 406 (20.4) 208 (10.4) 173 (8.7) 63 (3.2) 139 (7.0) 12 (0.6)
Severity of COVID-19 disease, n° (%) - Mild or Moderate - Severe - Critical - Unknown	784 (39.3) 348 (17.5) 577 (28.9) 284 (14.3)
SOFA score (median, IQR)	2 (3)
Supplemental oxygen therapy, n° (%) - None - Nasal cannula or simple face mask - HFNC - CPAP/NIV - Mechanical ventilation	260 (13.0) 823 (41.3) 159 (8.0) 370 (18.6) 17 (0.9)
Baseline laboratory parameter (mean, SD) - WBC (×1000/μL) - Lymphocytes count (×1000/μL) - INR - Creatinine (mg/dL) - ALT (UI/mL) - Total bilirubin (mg/dL) - PCR (× ULN) - PCT (ng/mL)	8.8 (6.6) 1.1 (2.3) 1.17 (0.54) 1.25 (1.44) 51.7 (100.3) 0.78 (1.2) 21.9 (93) 1.8 (8.7)
Corticosteroid treatment, n° (%)	1436 (72.1)
Antimicrobial treatment - Macrolides - Other antibiotic treatment	416 (20.8) 478 (24.0)
Suspected or confirmed bacterial infections, n° (%) - PCT ≥ 1 - Co-infection - Superinfection	267 (13.4) 17 (0.9) 25 (1.25)
Site of infection, n° (%) - Primary/CVC-related BSI - HAP/VAP - UTI - Intra-abdominal infection - Skin and soft tissue - Others	42 10 (23.8) 4 (9.5) 15 (35.7) 3 (7.1) 3 (7.1) 7 (16.7)
Etiology, n° (%) - *S. aureus* - *Enterococcus* spp. - Other Gram-positive bacteria - *Enterobacterales* - *P. aeruginosa* - *A. baumannii* - Other Gram-negative rods - *C. difficile*	7 (16.7) 9 (21.4) 7 (16.7) 12 (28.6) 3 (7.1) 2 (4.8) 1 (2.4) 1 (2.4)
Hospital mortality, n° (%)	207 (10.4)

ID: Infectious Diseases; COPD: Chronic Obstructive Pulmonary Disease; HFNC: high-flow nasal cannula; CPAP: continuous positive airway pressure; NIV: non-invasive ventilation; BSI: bloodstream Infection; HAP: Hospital-acquired Pneumonia; VAP: Ventilator-associated Pneumonia; UTI: Urinary tract infection.

**Table 2 antibiotics-12-01124-t002:** Demographic, clinical, and biochemical characteristics of patients stratified according to the presence of bacterial infection.

	Bacterial Infection	No Bacterial Infection	*p* Value
N° of patients	309	1684	
Mean age (SD), years	63.1 (17)	62.1 (15.8)	0.32
Males, n° (%)	194 (62.8)	1032 (61.3)	0.62
Mean length of hospital stay (SD), days	16.9 (10.0)	15.9 (10.6)	0.12
Charlson Comorbidity Index (median, range)	15.5 (12)	14 (12)	0.21
Comorbidities, n° (%) - Cardiovascular disease - Diabetes - COPD - Chronic kidney disease - Chronic liver disease - Malignancies - HIV	98 (31.7) 79 (25.6) 28 (9.1) 43 (13.9) 23 (7.4) 21 (6.8) 4 (1.3)	458 (27.2) 327 (19.4) 180 (10.7) 130 (7.7) 40 (2.4) 118 (7.0) 8 (0.5)	**0.028** **0.006** 0.12 **<0.001** **<0.001** 0.061 **<0.001**
Severity of COVID-19 disease, n° (%) - Mild or Moderate - Severe/Critical	152 (49.2) 157 (50.8)	632 (45.1) 768 (54.9)	0.19
SOFA score (median, IQR)	0 (1)	1 (1)	0.80
Supplemental oxygen therapy, n° (%) - None - Nasal cannula or simple face mask - HFNC - CPAP/NIV - Mechanical ventilation	51 (16.6) 184 (59.7) 33 (10.7) 35 (11.3) 5 (1.6)	209 (15.8) 639 (48.4) 126 (9.5) 335 (25.4) 12 (0.9)	**<0.001**
Baseline laboratory parameter (mean, SD) - WBC (×1000/μL) - Lymphocytes count (×1000/μL) - INR - Creatinine (mg/dl) - ALT (UI/mL) - Total bilirubin (mg/dl) - PCR (× ULN)	9.8 (12.7) 1.3 (4.9) 1.19 (0.4) 1.6 (1.9) 55.4 (98.6) 0.9 (1.19) 17.6 (68.2)	8.6 (3.8) 1.0 (0.7) 1.16 (0.6) 1.2 (1.3) 50.7 (100.8) 0.8 (1.19) 23.3 (99.3)	0.11 0.26 0.22 **<0.001** 0.46 0.21 0.29
Corticosteroid treatment, n° (%)	273 (88.3)	1164 (69.1)	**<0.001**
Antimicrobial treatment, n° (%) - Macrolides - Other antibiotic treatment	111 (36.0) 155 (50.2)	305 (18.1) 323 (19.2)	**<0.001** **<0.001**
Hospital mortality, n° (%)	33 (10.7)	174 (10.3)	0.85

COPD: Chronic Obstructive Pulmonary Disease; HFNC: high-flow nasal cannula; CPAP: continuous positive airway pressure; NIV: non-invasive ventilation; BSI: bloodstream Infection; HAP: Hospital-acquired Pneumonia; VAP: Ventilator-associated Pneumonia; UTI: Urinary tract infection. *p* values < 0.05 are displayed in bold.

**Table 3 antibiotics-12-01124-t003:** Demographic, clinical, and biochemical characteristics of patients who received or did not receive antimicrobial treatment other than macrolides.

	Antimicrobial Treatment	No Antimicrobial Treatment	*p* Value
N° of patients	478	1515	
Mean age (SD), years	65.2 (16)	61.3 (15.9)	**<0.001**
Males, n° (%)	306 (64.0)	920 (60.7)	0.19
Waves of pandemic, n° (%) - First - Second - Third	4 (0.8) 326 (68.2) 148 (31.0)	313 (20.7) 754 (49.8) 417 (27.5)	**<0.001**
Mean length of hospital stay (SD), days	16.6 (10.5)	15.8 (10.6)	0.17
Charlson Comorbidity Index (median, IQR)	3 (4)	2 (3)	**0.002**
Comorbidities, n° (%) - Cardiovascular disease - Diabetes - COPD - Chronic kidney disease - Chronic liver disease - Malignancies - HIV	148 (31.0) 117 (24.5) 50 (10.5) 59 (12.3) 22 (4.6) 42 (8.8) 3 (0.6)	408 (26.9) 289 (19.1) 158 (10.4) 114 (7.5) 41 (2.7) 96 (6.3) 9 (0.6)	0.14 **0.017** 0.93 **0.02** 0.09 0.08 0.97
SOFA score (median, IQR)	2 (3)	1 (2)	**<0.001**
Supplemental oxygen therapy, n° (%) - None - Nasal cannula or simple face mask - HFNC - CPAP/NIV - Mechanical ventilation	44 (9.2) 220 (46) 49 (10.2) 151 (31.6) 9 (1.9)	216 (18.7) 603 (52.2) 110 (9.5) 219 (18.9) 8 (0.7)	**<0.001**
COVID-19 severity, n° (%) - Mild/moderate - Severe/critical	167 (34.9) 311 (65.1)	617 (50.1) 614 (49.9)	**<0.001**
Baseline laboratory parameter (mean, SD) - WBC (×1000/μL) - Lymphocytes count (×1000/μL) - INR - Creatinine (mg/dL) - ALT (UI/mL) - Total bilirubin (mg/dL) - PCR (× ULN)	9.9 (6.4) 1.2 (4) 1.2 (0.7) 1.4 (1.5) 53.9 (83.9) 0.9 (1.9) 20.3 (87.6)	8.4 (6.7) 1.0 (0.7) 1.15 (0.5) 1.2 (1.4) 50.7 (106.9) 0.7 (0.7) 22.6 (95.3)	**<0.001** 0.38 0.09 0.07 0.54 0.053 0.69
Corticosteroid treatment, n° (%)	445 (93.1)	991 (65.4)	**<0.001**

COPD: Chronic Obstructive Pulmonary Disease; HFNC: high-flow nasal cannula; CPAP: continuous positive airway pressure; NIV: non-invasive ventilation; BSI: bloodstream Infection; HAP: Hospital-acquired Pneumonia; VAP: Ventilator-associated Pneumonia; UTI: Urinary tract infection. *p* values < 0.05 are displayed in bold.

**Table 4 antibiotics-12-01124-t004:** Independent predictors of bacterial infections among COVID-19 patients enrolled.

Variable	OR	LCI	UCI	*p* Value
Cardiovascular disease	1.49	0.51	4.24	0.47
COPD	1.71	0.31	9.43	0.53
Chronic kidney disease	1.05	0.16	6.99	0.96
Diabetes	2.78	0.91	8.47	0.07
Supplemental oxygen therapy (MV, NIV, CPAP, or HFNC vs. other)	0.71	0.20	2.5	0.59
Serum Creatinine	1.04	0.75	1.45	0.81
Corticosteroid treatment	2.34	0.63	8.66	0.20

LCI: Lower Confidence Interval; UCI: Upper Confidence Interval; COPD: Chronic Obstructive Pulmonary Disease; HFNC: high-flow nasal cannula; CPAP: continuous positive airway pressure; NIV: non-invasive ventilation; MV: mechanical ventilation.

**Table 5 antibiotics-12-01124-t005:** Independent predictors of antimicrobial prescription among COVID-19 patients enrolled.

Variable	OR	LCI	UCI	*p* Value
Age	1.007	0.997	1.018	0.176
Waves (third ref.) - First vs. third - Second vs. third	0.896 1.53	0.18 1.18	4.57 1.97	0.895 **0.001**
Diabetes	1.074	0.79	1.46	0.645
Chronic kidney disease	1.233	0.79	1.90	0.347
CCI (>3 vs. ≤3)	1.107	0.785	1.561	0.562
SOFA (>2 vs. ≤2)	2.05	1.53	2.75	**<0.001**
Supplemental oxygen therapy (MV, NIV, CPAP, or HFNC vs. other)	1.009	0.74	1.37	0.957
Severity (severe/critical vs. mild/moderate)	1.663	1.241	2.228	**0.001**
WBC count (×1000/μL)	1.03	1.004	1.06	**0.023**
Corticosteroid treatment	1.289	0.797	1.905	0.347

LCI: Lower Confidence Interval; UCI: Upper Confidence Interval; CCI: Charlson Comorbidity Index; HFNC: high-flow nasal cannula; CPAP: continuous positive airway pressure; NIV: non-invasive ventilation; MV: mechanical ventilation. *p* values < 0.05 are displayed in bold.

## Data Availability

The data presented in this study are available on request from the corresponding author.

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
