# Peer review of "Prevalence and Epidemiological and Clinical Features of Bacterial Infections in a Large Cohort of Patients Hospitalized for COVID-19 in Southern Italy: A Multicenter Study"

_antibiotics, 2023, doi:10.3390/antibiotics12071124_

Round 1

Reviewer 1 Report

Firstly, I would like to congratulate the authors on this vast study, my comments and suggestions are as follows.

You mention you used the CCI and SOFA indexes, why did you choose these in particular? I think this warrants a phrase as to why you chose these.

Lines 132 to 135 you mention the types of respiratory support the patients received during their admission; however, it is rather difficult to discern due to the way it is phrased, I think this should be expressed in a clearer manner due to the fact that it is a important element of the study.

You mention in lines 153 to 156 the various comorbidities such as cardiovascular disease and CKD yet you should also mention that COVID 19 also has serious effects on these systems since there are many studies that approach these subjects, you may refer to DOI: 10.1007/s40620-021-01206-8 for CKD and DOI10.3390/jpm11030225 and DOI10.3390/jcm10020199 for cardiovascular disease and others of your liking.

You have 284 patients that have “unknown” COVID 19 severity, shouldn’t these patients have been discarded from the study to begin with?

Since you mention the waves of the pandemic more than once (for example line 173), you should also discuss the fact that during the first wave of the pandemic patient admissions were much lower and this is an important factor to take into consideration. I think the authors should include a short paragraph mentioning the impact of the pandemic on hospitalizations, you may refer to 10.3390/medicina57050441 ; 10.1377/hlthaff.2020.00980 and 10.1371/journal.pone.0252441.

 Line 184 to 186 you state that subjects without superinfection had a higher prevalence of severe COVID 19 infection, this should be further detailed and corroborated with data from literature, please include adequate citations.

Line 207 these findings should be presented together with similar findings, citing literature and studies that have had similar conclusions.

 Lines 230 231 you mention “ {….}demonstrated a higher incidence of ventilator associated pneumonia in 230 the first group, not fully explained by the prolonged ventilation of patients “. The authors should elaborate based on adequate literature citing.

Rather good quality English, no special mentions here. 

Author Response

Point 1: You mention you used the CCI and SOFA indexes, why did you choose these in particular? I think this warrants a phrase as to why you chose these.

Answer: Following the suggestion of the reviewer, we added in the Methods section an explanation for this choice (lines 96-99).

Point 2 Lines 132 to 135 you mention the types of respiratory support the patients received during their admission; however, it is rather difficult to discern due to the way it is phrased, I think this should be expressed in a clearer manner due to the fact that it is a important element of the study.

Answer: As suggested by the reviewer, the sentence was rephrased to make it clearer (lines 150-152).

Point 3: You mention in lines 153 to 156 the various comorbidities such as cardiovascular disease and CKD yet you should also mention that COVID 19 also has serious effects on these systems since there are many studies that approach these subjects, you may refer to DOI: 10.1007/s40620-021-01206-8 for CKD and DOI10.3390/jpm11030225 and DOI10.3390/jcm10020199 for cardiovascular disease and others of your liking.

Answer: We added this point and the citations suggested by the reviewer (lines 273-277)

Point 4: You have 284 patients that have “unknown” COVID 19 severity, shouldn’t these patients have been discarded from the study to begin with?

Answer: Since the severity of COVID-19 was not the main outcome of our paper, we decided not to exclude these patients from the study; however, patients with unavailable data were excluded from the multivariate analysis.

Point 5: Since you mention the waves of the pandemic more than once (for example line 173), you should also discuss the fact that during the first wave of the pandemic patient admissions were much lower and this is an important factor to take into consideration. I think the authors should include a short paragraph mentioning the impact of the pandemic on hospitalizations, you may refer to 10.3390/medicina57050441; 10.1377/hlthaff.2020.00980 and 10.1371/journal.pone.0252441.

Answer: We added this point in the text (lines 55-63), with the references suggested by the reviewer.

Point 6: Line 184 to 186 you state that subjects without superinfection had a higher prevalence of severe COVID 19 infection, this should be further detailed and corroborated with data from literature, please include adequate citations.

Answer: As suggested by the reviewer, we added references supporting this point (See Vaughn CID 2021 and So, Intern Emerg Med 2022).

Point 7: Line 207 these findings should be presented together with similar findings, citing literature and studies that have had similar conclusions.

Answer: We modified the sentence (lines 243-245); studies reporting similar data (i.e. the correlation between antimicrobial prescription and clinical severity of patients) were reported in the discussion (lines 301-310)

Point 8 Lines 230 231 you mention “ {….}demonstrated a higher incidence of ventilator associated pneumonia in 230 the first group, not fully explained by the prolonged ventilation of patients “. The authors should elaborate based on adequate literature citing.

Answer: Following the suggestion of the reviewer, we modified the text accordingly.

Reviewer 2 Report

Thank you for giving me the opportunity to read and comment a report “Prevalence and Epidemiological and Clinical Features of Bacterial Infections in a Large Cohort of Patients Hospitalized for COVID-19 in Southern Italy: A Multicenter Study”, by Onorato L, et al.

The reviewed manuscript evaluated the prevalence of bacterial infections and the prevalence of antimicrobial prescriptions and identified independent predictors of both infections and antimicrobial prescriptions.

This paper is well written, correctly structured with a suitable research concept, the study limitations are addressed, and it is of relevance to readers of the journal. However, I have included a few comments for your consideration:

The introduction is rather short, comprising only two paragraphs. It is recommended for authors to enhance the contextualization of their research by providing additional details. For instance, a description of the study sites' contexts and the pertinent COVID-related issues would be beneficial.

Microorganism names should be formatted in italics (line 143).

Authors are required to provide explanations for all acronyms used in tables. These explanations should be placed at the bottom of each respective table in the manuscript.

Reviewer 3 Report

The article is very good and accurate, but corrections need to be made on it before publication: there are typographical and writing errors regarding the names of pathogens that should be written in italics throughout the text. Please explain about obtaining informed consent from patients in a retrospective study. In the case of the tables, all the summarized terms and abbreviations should be explained in the table's caption. The introduction of the article is written very briefly, which needs to be completed.

there are typographical and writing errors regarding the names of pathogens that should be written in italics throughout the text

Round 2

Reviewer 1 Report

Dear authors, the issues that I have pointed out have been mainly addressed, though not groundbreaking, your manuscript which is  well written and explores a rather important subject.

Overall, spelling and phrasing are fine, the phrases that were ambiguous have been corrected.